# Biology and Management of *Vulpia myuros*—An Emerging Weed Problem in No-Till Cropping Systems in Europe

**DOI:** 10.3390/plants9060715

**Published:** 2020-06-03

**Authors:** Muhammad Javaid Akhter, Peter Kryger Jensen, Solvejg Kopp Mathiassen, Bo Melander, Per Kudsk

**Affiliations:** Department of Agroecology, Aarhus University, DK-4200 Slagelse, Denmark; mohammadjaved@agro.au.dk (M.J.A.); pkj@agro.au.dk (P.K.J.); sma@agro.au.dk (S.K.M.); bo.melander@agro.au.dk (B.M.)

**Keywords:** life cycle, vernalization, seed dormancy, seed germination, seed longevity, cultural weed control, non-chemical weed control, chemical control, integrated weed management

## Abstract

Recently, *Vulpia myuros* has become a problematic grass weed species in parts of Europe. It is most common in no-till cropping systems. The inherent tolerance to several selective grass weed herbicides is of serious concern to the successful management of *V. myuros* in arable farming. Here, we reviewed the available knowledge about the biology of *V. myuros* to identify knowledge gaps and assess management efforts to identify best practices for control. *V. myuros* is a winter-annual species producing seeds with a short dormancy that can germinate at a wide range of conditions. Seed longevity in the soil is short. Little information is available on the influence of *V. myuros* on crop yield but some results suggest that yield losses can be significant. The findings provide a better understanding of the weedy characteristics of *V. myuros* and highlight that management strategies in Europe need to be diversified and integrate preventive and cultural control methods. Finally, we identify some of the management tools that should be considered to minimize the impact of *V. myuros* on European farming and future needs for research to develop sustainable integrated weed management strategies.

## 1. Introduction

The genus *Vulpia* comprises more than 20 species [1], of which several are considered to be weeds of agricultural crops. *Vulpia* spp. are C3 plants and are native to Central and Southern Europe and the Mediterranean area [2], but are widespread and have been reported on most continents. *Vulpia myuros* (L) Gmelin, known as rat’s-tail fescue, and *Vulpia bromoides* (L) Gmelin, known as barren or brome fescue, are the most common weeds in the genus and will be referred to collectively as *Vulpia* spp. The two species are impossible to distinguish at the early growth stages but, as they develop, they can be separated morphologically by differences in panicle and lower glume structure (Figure 1). Only *V. myuros* has been reported as a weed problem in Europe and therefore the focus of this review will be on this species.

Initially, *V. myuros* was not reported as a problematic weed in field crops, but its abundance has increased in cereal crops, particularly in North America and Australia, following the adoption of no-till cropping systems [2,3]. In Europe, *V. myuros* is a recent weed problem. The first occurrence in agricultural fields in Denmark was reported in red fescue (*Festuca rubra* L.) for seed production in the late 1990s [4]. A recent survey of grass weeds in grass seed crops revealed that *V. myuros* has become common in some regions of Denmark [5], and within the last five years, it has also been reported more frequently in winter cereals in Denmark and often at densities of several hundred plants/m^2^ [Melander, pers. com.]. *V. myuros* is also reported frequently in winter cereals in France [Bonin, pers. comm.]. Recently, *V. myuros* was also reported to be an upcoming weed problem in Romania in direct seeded winter oilseed rape [6], while in other countries, for example the UK, *V. myuros* is still considered a minor weed problem [7].

In recent years, the area under no-till cropping systems has increased throughout Europe [8]. This, in combination with an inherent tolerance to some of the most widely used graminicides [9], is likely the cause of increasing problems in both Denmark and Romania [6,8]. The inconsistent performance of herbicides, which provide consistent control of other important grass weed species, means that preventive and non-chemical control methods will have to be applied to prevent *V. myuros* from infesting fields.

The most recent reviews on *Vulpia* spp. date back to the 1990s [2,10]. Recent problems with *V. myuros* in parts of Europe call for an updated review summarizing the new information that has become available about this grass weed species. The objective of this review is to provide insight into the biology and ecology of *V. myuros*, including the impact on crop yield and quality, available control methods, and to translate this information into sustainable management strategies for European conditions. We also suggest directions for future research on the biology and management of *V. myuros* needed to refine management strategies. The review also includes results from studies on the closely related *V. bromoides* whenever relevant. In some countries, e.g., Australia, the two species often occur together [11], and when this is the case, the term *Vulpia* spp. will be used.

## 2. Biology

### 2.1. Botanical Description

*V. myuros* has a hexaploid genome with 42 chromosomes [1]. *Vulpia* species were previously classified under the genus *Festuca*, but moved to a genus of their own because their cleistogamous flowers, long-awned lemmas and annual growth habit contrasted with the typical *Festuca* attributes of chasmogamous florets, lemmas with short awns and a perennial growth habit [1].

Stem length of *V. myuros* varies considerably from 5 and up to 75 cm depending on the growing conditions. Leaf sheaths are overlapping and slightly hairy. Dense stands at early growth stages have a tufted appearance with slender, erect, and finely pointed leaves, with visible veins on the upper side. Leaf sheaths of *V. myuros* are split with few hairs. Inflorescences of *V. myuros* are 5 to 35 cm long, dense with green–purple color and 5 to 12 mm long spikelets with multiple florets per spikelet with 1 or 2 self-fertile stamens [1]. Seeds of *V. myuros* are 3.5 to 4.5 mm in length and 0.5 to 0.6 mg in weight [2,12,13]. Seeds of *V. myuros* attach easily to clothing, animal fur, wool and hairs [2,12,14]. The panicles are generally very narrow, slightly secondly positioned, partially enclosed in the leaf sheath and 5–35 cm long. The lower glume is 0.4–2.5 mm long, and is one tenth to two fifths the length of the upper glume.

### 2.2. Global Distribution and Habitat

*V. myuros* occurs as a weed of grasslands and annual crops in the world’s temperate regions [15]. *Vulpia* spp., particularly *V. bromoides*, are also weeds in Australia, North and South America, Northern India, South Africa, and Japan [2]. The causes for their expansion into new areas are currently poorly understood. Animals and humans are important dispersal vectors for transporting seeds over long distances [16], whilst abiotic factors, such as farm machinery, water and wind, are more important for seed spread among farms and fields.

Optimum conditions for growth appear to be cool and moist winters and warm summers, though without long dry spells [2]. *Vulpia* spp. occur on many soil types, ranging from clayey to sandy soils, with pH-values of 4.5 to 9 [17]. *V. myuros* is more successful as an arable weed than *V. bromoides*, because it germinates quickly under various environmental conditions, even from deeper soil layers. In addition, its initial vegetative growth rates are greater than those seen for *V. bromoides* [12].

### 2.3. Life Cycle

*V. myuros* mainly behaves as a winter annual weed [2], although early spring germination has been reported [18]. Its dormancy pattern and its ability to survive harsh winters as juvenile plants are important attributes for its ability to infest winter cereals. Growth cycle of *V. myuros* resembles that of winter wheat and dry matter accumulation typically follows a logistic growth pattern. Initial growth rate (first 7–10 weeks after emergence) varies according to temperature and, particularly, high temperatures (20–28 °C) and long photoperiods (16 or 12 h) during the initial growth phase result in vigorous growth [12]. Later (10–16 weeks after emergence), the growth rate declines or remains constant at high and intermediate temperatures (15–23 °C), whereas growth rate continues to increase exponentially with time at lower temperatures (10–18 °C). This adaptive response to temperature and photoperiod with age enables *V. myuros* to establish and grow quickly early in the growing season. As crop growth progresses, *V. myuros* becomes less competitive in contrast to other annual grass weeds, e.g., *Apera spica-venti* (L) P. Beauv [2,19,20].

Moderate temperatures (18/10 °C) and short days (8 h) promoted while high temperatures (28/20 °C) inhibited the initiation of flowering of *V. myuros* [12]. Flowering occurs typically in late June in Northern Europe and it takes about 18 days from flowering to seed maturity, although this can vary from 7 to 30 days, depending on growing conditions [21,22].

Under stress, *V. myuros* quickly shifts from the vegetative to the reproductive and into the generative phase and completes the life cycle faster, resulting in reduced fecundity and thus fewer new seeds entering the soil seed bank [12]. *V. myuros* is a shallowly rooted plant [23] and thus sensitive to drought conditions [24], but the ability to flower early under stress means that it can alleviate the detrimental effects of periods with water shortages [10].

### 2.4. Fecundity

*Vulpia* spp. are prolific seed producers and can establish a large soil seed bank [25]. The fecundity of *Vulpia* spp. is variable. Wallace [2] recorded 1–3.3 inflorescences per plant of *V. bromoides*, one inflorescence having 2.9–63.7 spikelets and one spikelet containing 0.8–4.4 caryopses. More than 200,000 seeds/m^2^ were produced in a stand of mixed pasture grasses with a 13% density of *V. bromoides* [11]. Time of emergence had a substantial impact on seed production. Under Australian conditions, plants emerging in March yielded 53 times more seeds than those emerging in July [21]. Plant fecundity of *V. myuros* in Australia was found to be greater than in France [26]. Upon dispersal, spikelets separate at the base of florets [1] and *Vulpia* spp. seeds disperse rapidly after maturity [2]. Their seeds are small and light and can easily attach to animals, machinery and clothing [2].

### 2.5. Seed Biology

#### 2.5.1. Vernalization Requirement

Vernalization is essential for the sexual reproduction and seed production of *V. myuros* [2]. Tarasoff et al. [18] studied the vernalization requirements of two populations of *V. myuros* collected in Oregon, USA, an eastern population from an area with a more continental climate (low temperatures during winter) and a western population from a more coastal climate with mild winters. They found significant differences in vernalization requirements between the two populations, supporting the hypothesis of vernalization plasticity. The eastern population responded more strongly to vernalization temperature than the western population, whereas both populations responded to vernalization length. Genetic variation or plasticity is an advantageous trait, as it offers species the possibility to adapt to changes in the environment and to establish itself in a new environment. Tarasoff et al. [18] also noticed that seeds produced on plants subjected to a long vernalization period had a higher germination rate than seeds produced on plants subjected to a shorter duration of vernalization. This indicates either more mature seeds or seeds with less dormancy in the newly harvested seeds from the plants subjected to long vernalization periods.

#### 2.5.2. Seed Dormancy

Most *Vulpia* spp. seeds germinate in the year of their production, with only 1–7% remaining dormant [2,21,27,28]. Seeds of *Vulpia* spp. need an after-ripening period of 2–3 months for germination [12]. Primary dormancy in newly shed *Vulpia* spp. seeds reduces germination in the summer, when no crop is present in the fields, and delays germination peaking to the autumn synchronized with the time of autumn established crops [10]. Dormancy and germination are influenced by the light environment. Green plant material reduces light intensity and quality at the soil surface. The fraction of dormant seeds increased from 1% in intensively grassed to 7% in lightly grassed pastures [21], as a response to the changes in the light environment. The significant inhibitory effect of crop biomass on the germination of *Vulpia* spp. was also noted by Peart [29]. When the seeds are able to sense changes in light quality caused by green biomass, they can avoid germination under conditions where competition from established plants are significant [30]. Ball et al. [31] reported that after-ripening and pre-chilling (5 days treatment at 5 °C) of the seeds were the primary factors reducing dormancy in *V. myuros* seeds.

#### 2.5.3. Seed Germination

Seed germination is a complex process depending on the conditions prevailing in the soil-seed interface, temperature, time of year and light conditions and extent and time of precipitation [10,31]. *V. myuros* can germinate over a wide range of temperatures and this range increases with increasing after-ripening time and when germinated in light [12]. Ball et al. [31] found a maximum germination rate of *V. myuros* at approximately 20 °C but, in general, the germination rate was high when the temperature ranged from 5 to 30 °C. Scherner et al. [32] studied the germination of a Danish population of *V. myuros* and found that the base temperature for germination was 1 °C. A thermal time of 92 to 111 degree days was required to reach 50% germination estimated from germination tests made in darkness over a wide range of constant temperatures. The model predicts 5 days to reach 50% germination at 20 °C, while 13 days are required at 7 °C. Light promoted *Vulpia* spp. germination and made it more uniform over a range of temperatures [21]. Maximum time for 50% germination at 22.5 °C was found to be around 1.5 days in light, compared with three days in darkness for *V. myuros* [12].

The study by Scherner et al. [32] showed that germination was hampered by moisture contents lower than −0.1 MPa, and the effect of moisture became very apparent when the level was lower than −0.25 MPa. The germination of *V. myuros* over a broad range of temperature, light and pH regimes [12,33] suggest that seeds can germinate in every season, if the requirements for an after-ripening period and sufficient moisture are met.

#### 2.5.4. Seed Longevity and Seedling Establishment

Jensen [34,35] studied the survival of *V. myuros* seeds buried at different depths and for different intervals under field conditions over a 7-year period. There was no clear relationship between seed survival and soil depth for seeds incorporated at depths from 2 to 25 cm. However, seed survival of incorporated seeds was greater than for seeds placed on the soil surface. The percentage of viable seeds was reduced to less than 8.5% after just 1-2 months at the soil surface. Although very variable over years, the percentage of viable seeds after one year of incorporation was reduced to 0.1 to 50% at 10 cm soil depth, indicating a short seed longevity. The results align with the findings of Dillon and Forcella [12], that *V. myuros* can be found in fields with conventional tillage, where at least a small proportion of the seeds have survived in the soil for a period of one year.

Seedling recruitment of *Vulpia* spp. is mainly dependent on the annual seed rain [10], because of the short seed longevity in soil [27,31]. One season without seed production decreased seedling density of *V. bromoides* in grassland to 28 plants m^−2^, compared to 904 plants m^−2^ in treatments where seed shedding was not prevented [29].

## 3. Impact on Crop Yield

Often, *V. myuros* invade from field edges [2], or infestation begins in bare patches in the field [22]. Once established, *V. myuros* is very competitive in winter cereals during the early crop growth phases, partly because of its ability to establish turf-like stands under warm weather conditions. High infestations of *V. myuros* can create thick mats and form a physical barrier; hence, dead *V. myuros residues* from a previous crop can hinder proper operation of the sowing machine in no tillage systems [36], and cause establishment failure of a succeeding autumn sown crop [37].

Some publications reported an impact of *V. myuros* on crop growth and yields, including winter wheat, winter oilseed rape and red fescue [4,6,38,39] but information on quantitative yield losses is scarce. Lawrence and Burke [39] found yield reductions in winter wheat in Washington, US of 37 to 45% at *V. myuros* biomasses of 135 to 202 g m^−2^ measured close to grain harvest, while Ball et al. [38] found that crop yield losses varied significantly between years and sites. There are no studies examining the interference of *V. myuros* on growth and yield of winter cereals under Northern European conditions, hence, it is not possible to compare the impact of *V. myuros* to other important grass weeds like *A. spica-venti, Alopecuros myosuroides* Huds. and *Lolium multiflorum* Lam.

*V. myuros* produces allelochemicals that suppress the growth of other plants. In glasshouse studies, residues of *V. myuros* significantly suppressed germination and seedling growth of other plant species, including both crop and weed species [40,41]. Yamamoto and Kato-Noguchi [41] found that water extract and powder of *V. myuros* were equally effective in suppressing the root and shoot growth of *Lepidium sativum*. Twenty-one phytotoxic compounds were identified from aqueous extract of *V. myuros*, accounting for 0.05% of the dry weight residue [42]. The biological activity of the majority of the compounds was low to medium and the allelopathic effect was caused by the combined effect of the compounds rather than a few of the 21 compounds [43]. Phytotoxic effects have also been verified under field conditions; winter wheat yield was reduced by up to 66% following the application of *Vulpia* spp. residues at a rate of 1 t ha^−1^ [44].

## 4. Management

### 4.1. Cultural Weed Control

The main objectives of cultural weed control are to reduce weed densities and to diminish weed crop competition by improving crop competitiveness [45]. *V. myuros* has seeds with limited longevity in soil and this characteristic can be used in different ways to reduce infestations. Periodic inversion by moldboard ploughing was an effective tactic to reduce the population of *V. myuros* compared to non-inversion tillage [46]. Leaving the seed undisturbed on the soil surface in the period between two crops is also effective in reducing the *V. myuros* seed bank [34,35]. During cropping, higher seed rates and adequate amounts of fertilizers can enhance the crop suppression of *V. myuros* [2].

The vernalization requirement suggests that crop rotation can have a major influence on populations of *V. myuros*. A seed bank study showed that continuous cropping of winter wheat promoted *V. myuros*, while the inclusion of spring-sown crops in crop rotations interrupted the life cycle of *V. myuros* and prevented seed production [46].

### 4.2. Physical Weed Control

There is no specific information available on the effectiveness of physical methods applied directly in annual field crops against *V. myuros*. However, annual grass species tend to respond similarly to mechanical and thermal methods in terms of sensitivity in relation to growth stage. Imbedded grass seeds in the soil are easily destroyed using soil steaming techniques, provided that the maximum soil temperature reaches a minimum of 70 °C [47,48]. Hot water application and flaming can also control annual grasses at young growth stages by killing the aboveground plant tissue [48]. Flaming is very sensitive to growth stage and its effectiveness declines rapidly as grasses develop. Thus, repeated treatments and high gas doses are required for satisfactory control [47]. Soil solarisation may also be an effective thermal method under conditions with high solar radiation, as was demonstrated recently for the grass species *Bromus tectorum* (L) [49]. Due to low capacity in terms of area covered per hour and/or high costs, thermal methods only have relevance in amenity areas and high value crops.

Weed harrowing with flex tine harrows has limited effect on annual grasses unless treatments are targeted at very juvenile growth stages, preferably the white thread stage [8,50]. Mechanical weed control tools with a cutting action, such as shares, are far more effective for grass weed control and less sensitive to growth stage at the time of application [51]. Shares can be mounted on inter-row cultivators and inter-row hoeing in organic cereals is becoming more popular [52]. The hoe controls weeds in the inter-row zones effectively, while the crop itself suppresses intra-row weeds. Increased seeding rates, competitive cultivars and the placement of fertilizers below the crop seeds are all factors that improve this suppression [51].

Mechanical defoliation is an effective management technique in pastures, especially when deployed before seed shedding [53]. Grazing also defoliates aboveground vegetation and Dowling et al. [54] obtained significant reduction in the regeneration of *Vulpia* spp. following grazing of an Australian pasture when 90% of the tillers had elongated. Finally, hand weeding has been suggested where small infestations occur and in sensitive habitats, like ephemeral pools [55].

### 4.3. Chemical Weed Control

#### 4.3.1. Selective Herbicides

Early studies on the chemical weed control of *V. myuros* involved primarily residual herbicides. Lee [56] reported the very effective control of *V. myuros* with ethofumesate applied pre- or early post-emergence to a *L. multiflorum* seed crop, but the dose required to control *V. myuros* was ca. 100% higher than the dose needed to control *Poa annua* (L). Mueller-Warrant and Niedlinger [57] achieved full control of *V. myuros* with oxyflourfen and pendimethalin applied pre-emergence and at the 2-leaf stage of the weed, but applied at later growth stages, oxyflourfen was less effective. Ball et al. [38] found higher and less variable effects against *V. myuros* of flufenacet (94–100%) than of pendimethalin (10–99%) in imidazolinone tolerant winter wheat. Pre-emergence application of pyroxasulfone provided more than 74% control of *V. myuros* and was comparable to flufenacet, while pyroxsulam did not provide consistent control [39]. Applied pre-incorporated in winter wheat, prosulfocarb + s-metolachlor were more effective (82–83%) against *V. myuros* than trifluralin (72–76%) [58].

Sequential treatments, combining a pre-emergence treatment with a post-emergence application of an acetolactate synthase (ALS) inhibitor, were generally more effective and performance was less variable than applications of only pre-emergence herbicides [22,39]. In winter wheat, pendimethalin applied pre-emergence followed by iodosulfuron + mesosulfuron early post-emergence (3 weeks later), resulted in a 96% reduction in the density of *V. myuros*, while a tank mixture of the two herbicides applied early post-emergence only provided 82.5% control [22].

#### 4.3.2. Non-Selective Herbicide

The effective control of *Vulpia* spp. was also reported with pre-sowing/pre-emergence treatments with non-selective herbicides like glyphosate, paraquat, simazine and diuron [59,60,61]. Higher effects of glyphosate compared to diquat were reported in herbicide screenings in pot as well as field experiment [62]. Sequential applications of glyphosate was a good option and more effective than a single application against *V. myuros* in a chemical-fallow cropping system, as was glyphosate in combination with paraquat + diuron [36]. In grassland, the application of glyphosate, simazine and paraquat at sub-lethal rates around panicle emergence in spring, referred to as spray-topping, reduced the *Vulpia* spp. population and seed production in the succeeding crop [11,63]. Diquat, paraquat and simazine are no longer authorized in the EU, i.e., glyphosate is the only non-selective herbicide left on the market and it is currently up for re-authorization [64].

#### 4.3.3. Natural Tolerance to ACCase and ALS Inhibiting Herbicides

Yu et al. [9] reported that a range of acetyl-coenzyme A carboxylase (ACCase) herbicides and the ALS inhibiting herbicide chlorsulfuron were ineffective against *V. bromoides*. They found that tolerance to ACCase inhibiting herbicides was due to an insensitive target ACCase. A similar mechanism of tolerance against ACCase inhibiting herbicides was reported in several closely related *Festuca* spp. [65,66,67]. The mechanism of tolerance to ALS herbicides was found to be due to an elevated capacity for cytochrome P450 catalyzed metabolism [9]. More recent studies have shown, however, that some ALS inhibitors, such as pyroxsulam, mesosulfuron, propoxycarbazone and sulfosulfuron, can provide some level of control of *V. myuros* when applied at early growth stages [4,7,39]. No studies have compared the susceptibility of *V. myuros* and *V. bromoides*, but the available results indicate that their responses to herbicides are very similar. Thus, it can be assumed that the tolerance of *V. myuros* to ACCase and ALS inhibiting herbicides is caused by the same mechanisms as reported for *V. bromoides*.

Due to the natural tolerance to ACCase inhibitors and the unsatisfactory performance of many ALS inhibitors, residual herbicides with different sites of action, like flufenacet, pendimethalin and prosulfocarb, will likely be the mainstay of any chemical management strategy of *V. myuros* [7].

#### 4.3.4. Weed Resistance

The chemical control of many grass weeds is often hampered by resistance to the most widely used herbicides [68]. *V. myuros* is among the grass weeds for which weed resistance is not yet a major issue. This can probably be attributed to the natural tolerance to ACCase and ALS inhibitors, which have discouraged farmers from using two of the most resistance prone herbicide groups [67]. Until now, only two cases have been reported, both in Australia. Paraquat resistance was confirmed in *V. bromoides*, with the LD_50_ of resistant plants of *V. bromoides* being five- to six-fold higher than of the susceptible biotype [69]. Recently, Ashworth et al. [3] reported a case of simazine resistance in *V. bromoides*. The resistant biotype was > 596-fold more tolerant to simazine than a susceptible biotype. Continued use of triazine herbicides (8 times within 12 years) was assumed to be the cause of resistance. Hull et al. [7] tested the susceptibility of nine *V. myuros* populations from Denmark and the UK to three residual herbicides, flufenacet, pendimethalin and prosulfocarb with different sites of action, and found no indication of resistance. Because of the natural resistance to ACCase and ALS inhibitors, glyphosate application in between crops has become an integrated part of strategy to control *V. myuros* and this may result in the evolution of resistance to glyphosate, as has been observed for *Bromus sterilis* (L.) in the UK [70].

## 5. Perspectives for *V. myuros* Management in Europe

Although *Vulpia* spp. have been reported as serious weed species in pastures and annual crops in Australia since the 1980s [12,71], it is only within the last 10 years reports suggesting *V. myuros* as a weed problem in Europe have emerged. Although studies are available on the biology and management of *Vulpia* spp., the information is not always relevant in a European context, due to different climatic conditions, growing seasons and cropping systems. For example, problems with *Vulpia* spp. in arable crops in Australia are primarily caused by the prevalence of *Vulpia* spp. in pastures grown in rotation with winter annual crops like winter wheat and triticale, which is not a common crop sequence in Europe.

From a management point of view, the most challenging feature of *V. myuros* is its inherent tolerance to ACCase herbicides and low susceptibility to many ALS inhibitors [9]. Hence, farmers are left with fewer chemical options and are more reliant on residual herbicides than for the control of other important grass weed species in annual crops. Residual herbicides have a narrower window of application (pre- to early post-emergence) and often display a more variable efficacy, because their performance is influenced by soil moisture and soil organic matter content [72]. Furthermore, the majority of the currently available residual herbicides were introduced to the market many years ago and several are in risk of either being banned or have their use restricted, due to the strict EU pesticide legislation [73]. In addition, *V. myuros* has been found to tolerate typical use rates of glyphosate better than other grass weeds [36,38], which is a problem, particularly in no-till and direct seeded cropping systems where grass weed control depends on the use of glyphosate. Whether this is also caused by natural tolerance or other factors, e.g., low spray retention on the erect and narrow leaves of *V. myuros*, is at present unknown. Additionally, physical control methods are currently not an alternative for *V. myuros* control, due to either low effectiveness, low capacity or high costs. Thus, for the effective management of *V. myuros* in European arable farming, it is imperative to identify and develop preventive and cultural control measures that, applied in combination, can constitute future sustainable integrated weed management strategies. Based on the outcome of this review, the most promising methods emerging are crop rotation, soil cultivation, seedbed preparation, increased crop competition and weed seed harvest.

### 5.1. Crop Rotation

*V. myuros* requires vernalization, hence the inclusion of spring-sown crops in the crop sequence should be a very effective measure. Recently, this presumption was substantiated by Scherner et al. [46], who found that the inclusion of spring crops in a 12-years crop rotation dominated by winter cereals significantly reduced the number of *V. myuros* seeds accumulated in the seed bank. The effect of spring cropping on the occurrence of *V. myuros* was more pronounced than for *A. spica-venti*, another common grass weed in winter cereals in Northern and Central Europe. Apart from this study, no studies have looked specifically on the effect of spring-sown crops on the population dynamic of *V. myuros*, and the study by Scherner et al. [46] did not allow for any conclusions on the number of years with spring crops required to prevent the build-up of a *V. myuros* population. Seed biology studies have revealed a lower seed longevity of *V. myuros* compared to *A. myosuroides* [34] and *A. spica-venti* [74] and the observation by Scherner et al. [46], that the inclusion of spring crops had more effect on the presence of *V. myuros* than *A. spica-venti*, substantiate these findings. Nonetheless, crop rotation studies are required to validate these assumptions and provide more precise guidance on the effects of crop rotation.

If vernalization requirements vary between European populations of *V. myuros*, as was found in the US [18], plants germinating in early sown spring crops may be able to produce seeds. Although seed production of these plants will probably be much lower than of autumn germinating individuals, they could potentially maintain the soil seed bank and reduce the effect of crop rotation as a control measure. No information is available on the vernalization requirements of European *V. myuros* populations, but this information is needed to develop robust guidelines on crop rotation practices. In summary, crop rotation appears to be not only a very effective measure to prevent the build-up of *V. myuros* populations, but also an effective method to resolve an existing problem with *V. myuros*. However, more results from long-term trials are required to develop crop specific crop rotation guidelines.

### 5.2. Soil Cultivation

A recent study comparing ploughing, harrowing to 8-10 cm depth and direct seeding suggested that *V. myuros* might respond more strongly to both ploughing and harrowing than *A. spica-venti* [46]. Ball et al. [31] also noticed that *V. myuros* was most abundant in direct-seed cropping systems with minimal soil disturbance. The pronounced effect of soil cultivation on *V. myuros* can be explained by the short seed longevity when buried in the soil [31,34,35]. Although soil cultivation and moldboard ploughing, in particular, is an effective measure against *V. myuros* and other yield-reducing grass weeds with short seed longevity [75], non-inversion tillage and no-till is being promoted in many European countries as a measure to preserve soil fertility and reduce soil erosion [8]. In no-till systems, occasional ploughing, also referred to as strategic tillage, could be a solution to prevent the build-up of large, yield-reducing *V. myuros* populations [76], but this may, on the other hand, reduce the perceived benefits of no-till at least in the short-term [77], and therefore not be acceptable to no-till farmers.

### 5.3. Seedbed Preparation

Primary seed dormancy in *V. myuros* is short and the majority of the seeds germinate in the year of their production. Jensen [35] found that leaving seeds of *V. myuros* on the soil surface for 1–2 months reduced the density of *V. myuros* by 80%. Hence, the delayed sowing of winter annual crops could be an effective tool in an integrated weed management strategy against *V. myuros* in winter annual crops. Incorporating the seeds just a few cm into the soil increased seed persistence [35], i.e., preparing a false or stale seedbed in the autumn may be less effective than just leaving the seeds on the surface and postpone sowing. This technique is very well suited to no-till cropping systems. The risk associated with delayed sowing under the humid Northern European conditions is that a wet autumn may result in a poor seedbed or even force the farmer to abandon sowing due to wet soil conditions. Seed dormancy can vary between years. Swain et al. [78] found that low temperatures during seed maturation led to longer seed dormancy in *A. myosuroides*, which will reduce the effect of delayed sowing. A preliminary study on the influence of temperature during seed development did not suggest a relationship between temperature during seed formation and maturation and seed dormancy for *V. myuros* (Jensen, pers. comm.).

### 5.4. Crop Competition

*Vulpia* spp. have been reported to be less competitive than *Lolium rigidum* Gaudin and *Bromus* spp. and to reduce crop yields only at high densities [16,79], however, this conclusion emerged from studies conducted under conditions not comparable to those of Northern Europe. Studies conducted in the US Pacific Northwest revealed that yield losses in winter wheat could reach almost 50% [38,39] and although no data is available on the potential yield losses inflicted by *V. myuros* under European cropping conditions, they can be expected to be significant. Limiting yield losses can be achieved by increasing the competitive ability of the crop and/or weakening the competitiveness of the weed. Sowing time, sowing density, sowing pattern and competitive crops/cultivars are tools that can reduce the competition for resources [48,70,80,81]. Applied alone, the weed suppressing effect of any of these measures may be minimal, but in combination, high levels of control can be attained, because they work in a complementary fashion [82]. No results are available on the impact of these measures on the competitive ability of *V*. *myuros* and there is an imminent need to generate more information on the effects of these cultural practices on *V. myuros* impact on crop yields. Information available on other winter-annual grass weed species like *A. spica-venti* and *A. myosuroides* suggest that drilling date and sowing density would be effective measures to minimise the impact on crop yield and fecundity [20,75].

### 5.5. Weed Seed Harvest

Recently, the harvesting of weed seeds has received attention as a method for reducing weed densities. Originally, the method was developed in Australia for the control of *L. rigidum* in winter wheat, where it is now widely used and has become an important part of integrated weed management strategies [83]. As with other annual grass weed species producing seeds with short seed longevity in soil, seedling recruitment of *V. myuros* depends very much on an annual seed rain [10]. The successful implementation of weed seed harvest in Australia can be attributed to the fact that a large fraction of *L. rigidum* seeds is retained at crop maturity [84]. Visual observations indicate that *V. myuros* sheds the majority of its seeds several weeks before winter wheat maturity, as was recently observed for *A. spica-venti* and *A. myosuroides* [85]. Thus, the weed seed harvest method may not be as effective as for *L. rigidum*, but it may contribute to the long-term control of *V. myuros* and serve as a last resort in fields where other control measures have failed. No data are available on seed retaining in *V. myuros* at crop maturity and this information is needed to assess the potential benefits of weed seed harvest technologies.

## 6. Conclusions

*V. myuros* has many of the same characteristics as other important winter annual grass weeds and some of the lessons learned from management of these weed species can be applied to *V. myuros*, but differences do exist. With access to only a few effective herbicides, developing and implementing integrated weed management strategies to prevent further spread of *V. myuros* is needed to sustain future crop production. Several preventive and cultural control methods are available to farmers, but the lack of information on certain aspects of the biology of this new weed species and its potential impact on crop yields could hamper the development of cost-effective integrated weed control strategies. In the long-term, effective and affordable physical weed control methods, such as camera-guided inter-row cultivators, in combination with cultural control measures to improve intra-row crop competition, may provide farmers with effective control options.

## Figures and Tables

**Figure 1 plants-09-00715-f001:**
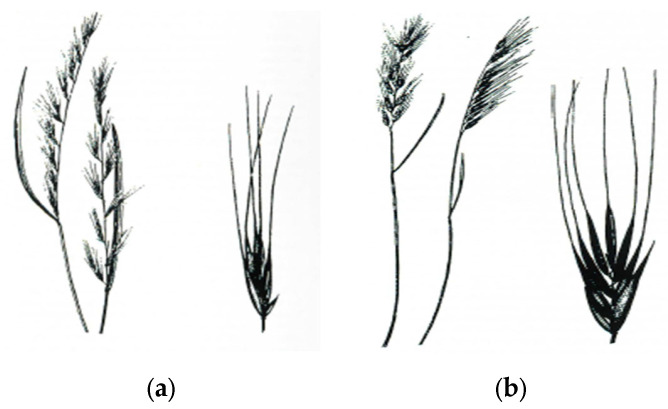
Panicle and spikelet of (**a**) *V. myuros* and (**b**) *V. bromoides*.

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
