# Peer review of "Biology and Management of Vulpia myuros—An Emerging Weed Problem in No-Till Cropping Systems in Europe"

_plants, 2020, doi:10.3390/plants9060715_

Round 1
Reviewer 1 Report
The manuscript deal with the biology and management of Vulpia myuros (L) Gmelin, an emerging weed in some European countries.
The review is well written, all the information reported are supported by the literature. In some cases external references would be welcomed.
Please see the attached document containing the specific comments regarding the diverse sections of the manuscript.
To my opinion, some variations are required. I guess they will improve the general quality of the paper.

Author Response
Response to comments by reviewer #1
Line 29: Please insert the name of the first scientist who classified the two species. Eg. Vulpia myuros (L) Gmelin, Vulpia bromoides (L) Gray
Done
Line 29: Move the sentence at line 30 (Vulpia spp. are…) to line 29 after the first sentence.
Done
Line 39: What about the spontaneous presence of the weed in southern European countries (France, Italy)? Is it a threat in these areas?
We have added a sentence concerning France where V. myuros has been observed in no-till systems (L. 62 in the track-change document).
Line 43: winter cerealS?
Corrected
Line 43: All the contents of this sentence refer to the same literature source (5)? If yes, please report it at the end of the sentence, otherwise insert a specific reference for the last part of the sentence.
No it does not. Melander (pers. comm.) was added as a reference to the last part of the sentence.
Line 46: What about Southern Europe?
See comment L. 39.
Line 47-49: Suggestion. Separate this long sentence in two ones. Suggestion: In the last XX ears (decades,…), the area covered ……[…] Europe. Stop the sentence here. This fact, in combination…..[…] Romania.
Corrected according to the suggestion by the reviewer.
Line 49: Revise the name of the state
Done.
Line 60-61: I understand what you wrote, but If you use Vulpia spp. you refer to all Vulpia species, and not exclusively to the two species under the focus. Please specify well that in this manuscript when the form Vulpia spp. is used referring to the two species previously mentioned in the text.
A sentence was added to clarify that Vulpia spp. refers to V. myuros and V. bromoides (L 47 in the track-change document).
Line 103: The flowering of Vulpia myuros is promoted by moderate temperatures (…..) and short daylight (8h), while high temperatures (….) shown an inhibitory action on flowering. What do you think? Sounds it better?
Corrected according to the suggestion by the reviewer. Yes, it sounds better.
Line 108: add a comma after faster.
Done.
Line 43 and Line 116: verify the apex (-2 or 2) of the SI metric measure used. In both case you talk about density
We checked the rules for Si measures and it should be /m2.
Line 117: Invert the sentence. Under Australian conditions, ……
Done
Line 121: …. and can be easily transported by animals, machinery and clothing.
Comma deleted
Line 170: Is the reduced seed longevity due to seed predation, biotic attack, etc? If there are some information about seed predation you can add a specific sub-section in section 5
We are not aware of any information on the causes of the short seed longevity. It is common phenomena for many grass weeds.
Line 180: ……is mainly dependent on the annual seed release (or deposition)
We prefer the expression ‘annual seed rain’.
Line 181-183: This is a twisted sentence. Please make it more clear.
The sentence was rephrased.
Line 189: hence dead V. myuros residues resulting from its development in a previous crop can cause…
The sentence was corrected to make it more clear.
Line 198: Please add the name of the first scientist who classified the last two reported species. Same thing for A. spica-venti at line 102.
Corrected.
Line 203-205: Which are the most allelopathic compounds? Is it possible to cite the most important ones? Are they clearly known or there is a cocktail effect?
A sentence and a reference was added to provided further details on the allelopathic effects of V. myuros.
Line 212-213: Hermetic sentence. Which are these strategies?
The sentence was rephrased. Should be more clear now.
Line 216: Please add the reference, unless it is the same of the succeeding sentences.
This sentence reflects the considerations of the authors, i.e. no reference.
Line 218: Please, add something about the potential effect of cover crops on Vulpia development.
We did not find any references looking into the effect of cover crops on V. myuros, hence we did not discuss this aspects of IWM.
Line 220-229: Please add other references (external references). The present document is a review for all scientists, you must consider other possible techniques even if they are unusual in Northern Europe (eg. solarization).
We added a sentence about soil solarization. We are aware that this review should cover all parts of Europe but there is no publications dealing with control methods applicable e.g. in Southern Europe.
Line 230: Please put mechanical treatments in a separate section. You have already mentioned the effect of mouldboard ploughing. Ploughing must be considered as both an agronomic and a mechanical method. Say something about spring-tooth harrowing (with elastic teeth >> see the picture). This kind of harrowing is very effective on weeds in winter cereals (when environmental and soil conditions are ok).
We disagree about the effectiveness of weed harrows against grass weeds. There is no evidence of any specific advantages using weed harrows against grass weeds. Results have been poor both from our own experiences and from the literature. We have changed the sentence on line 230 slightly.
We prefer to have a section on Physical methods rather than splitting it up into several sections.
Line 245-248: See the comment at Line 29 and Line 198 for Poa annua and L. multiflorum
Corrected.
Line 264: As the manuscript is reporting an emerging weed in Europe, please specify that diquat, paraquat and simazine are not allowed in Europe (not included in Annex I of Reg. 1107/2009). In addition, make some comments about the potential risk of appearance of Vulpia myuros resistant populations that the repeated use of glyphosate may comport. Any valid alternative to the expired herbicides? I am thinking about stale-seedbed [cultural and mechanical/chemical strategy (weeds may be killed by means of harrowing or with a non-selective herbicide)].
We added a sentence that the three herbicides are no longer authorized for use in the EU. Concerning glyphosate and the risk of resistance, we added a sentence and a reference in the section on weed resistance.
Line 300: it may be useful indicating which are the active ingredients or in alternative the chemical families involved.
The active ingredients were listed but not their mode of action.
Line 383. Wheat and barley hybrids. Which is the potential impact of these “new” varieties on Vulpia myuros development? Wheat and barley hybrids are more competitive, but they also leave more soil free for weed development due to the lower seed densities…. May Vulpia be easily controlled in hybrids due to crop competition?
We did mention ‘competitive crops/cultivars’ as a tool to minimize competition for resources. We have no specific information on hybrids but we believe this sentence also covers hybrids if they are more competitive.
Line 402-409: see comments on Line 245-248
Corrected.
Reviewer 2 Report
Akhter et al., Biology and management of Vulpia myuros – an emerging weed problem in no-till cropping systems in Europe
This paper provides a thorough review of the available literature on the biology of Vulpia myuros and suggested management strategies to reduce yield losses caused by infestations of Vulpia myuros. The paper addresses at length the life cycle of the weed, the conditions necessary for seed production, seed longevity, breaking seed dormancy and seed germination and how they can be exploited for management of Vulpia myuros. It very well summarizes the weed control strategies, including physical, chemical and cultural tactics that have been adopted / suggested in areas where Vulpia myuros has been found to be a problematic weed and how they translate into the European agricultural production systems. The review very clearly emphasizes on the need for research on the impacts of Vulpia myuros and its management under European cropping conditions for development of cost-effective integrated weed management strategies.
The manuscript, in general, is well written with appropriate citations. I think some sentences can be restructured for conciseness and clarity. While the paper mostly reads well as it is, I suggest a slight restructuring. Beginning on line 301 it is indicated that the rest of the paper will be about application to management in Europe; however, really it is just more review that follows. It seems like it would be better to complete the review above and then have a shorter section about application to Europe.
Comments by line:
47 The area “under” instead of “covered by”
104 “promote” rather than “promoted”
104 This is not clear. If short days produce flowering, why do they flower in June in northern Europe, when day length must be ~18 hr.
133, 134, 136 “subjected” and not “subject”
144 – 148 This bit of the paragraph is a little confusing to me. Line 144 suggests that the fraction of dormant seeds is lower (1%) under dense green biomass (lower intensity of light), indicating higher germinability. This is in contrast with line 147, stating that on sensing changes in light quality caused by green biomass, seeds avoid germination. Also, how can crop biomass have a significant inhibitory effect on both dormancy and germination? Shouldn’t it inhibit germination while promoting dormancy?
157 temperature “ranged from”
179 Add period after “year”
209 Don’t start sentence with abbreviated scientific name; spell out Vulpia.
211 Mouldboard plowing is a physical weed control method and should be included in physical section below.
215-218 Not clear what vernalization has to do with this. Please clarify.
236 Please expand upon what you mean by “placement of fertilizers.” Placement where?
237 This hardly seems worth mentioning in a review specifically about V. myuros.
268-270 This section about a grass sward is not clear. Was this a pasture? But followed by triticale?
302 Can replace “whereas” with “Even though/although”
303 The sentence structure does not sound right. How about “it is only within the last 10 years that the reports suggesting V. myuros as a weed problem in Europe have emerged.”
313-315 This sentence is a bit awkward. Please re-write for clarity.
322-323 Is this because the cropping systems typically don't use any type of physical or mechanical control? You provided evidence earlier that some cultivation can be successful. This needs clarification.
330 Maybe it is because I am not very familiar with the cropping systems in Europe, but I am still not clear on how V. myuros vernalization affects management. Please elaborate.
350-352 Elaborate. Is this because it allows different herbicides and different timing?
Author Response
Response to comments by reviewer #2
47 The area “under” instead of “covered by”
Corrected
104 “promote” rather than “promoted”
Corrected
104 This is not clear. If short days produce flowering, why do they flower in June in northern Europe, when day length must be ~18 hr.
The sentence was changed to highlight that this was the requirement for initiation of flowering.
133, 134, 136 “subjected” and not “subject”
Corrected
144 – 148 This bit of the paragraph is a little confusing to me. Line 144 suggests that the fraction of dormant seeds is lower (1%) under dense green biomass (lower intensity of light), indicating higher germinability. This is in contrast with line 147, stating that on sensing changes in light quality caused by green biomass, seeds avoid germination. Also, how can crop biomass have a significant inhibitory effect on both dormancy and germination? Shouldn’t it inhibit germination while promoting dormancy?
Correct. The words ‘dormancy and’ was deleted to clarify the relationship between light quality and dormancy.
157 temperature “ranged from”
Corrected.
179 Add period after “year”
‘Period of one’ was added before ‘year’.
209 Don’t start sentencWee with abbreviated scientific name; spell out Vulpia.
Corrected.
211 Mouldboard plowing is a physical weed control method and should be included in physical section below.
We disagree. Mouldboard ploughing is part of the cultivation technique and serves several purpose including incorporation of plant residues and is not a specific weed control measure. Hence, we think it should be included in the section on ‘cultural weed control’.
215-218 Not clear what vernalization has to do with this. Please clarify.
The fact that low temperatures and short days are required to initiate flowering (L 104) suggest that vernalization is required which led us to believe that crop rotation, in particular shifting between summer and winter annual crops could be an important cultural weed control methods. This assumption was confirmed by Scherner et al. (2016).
236 Please expand upon what you mean by “placement of fertilizers.” Placement where?
Further details provided.
237 This hardly seems worth mentioning in a review specifically about V. myuros.
Sentence deleted.
268-270 This section about a grass sward is not clear. Was this a pasture? But followed by triticale?
The sentence was deleted and partly incorporated in the following sentence to avoid repetition.
302 Can replace “whereas” with “Even though/although”
‘Whereas’ was relaced by ‘Although’.
303 The sentence structure does ‘not sound right. How about “it is only within the last 10 years that the reports suggesting V. myuros as a weed problem in Europe have emerged.”
The sentence was changed to improve clarity.
313-315 This sentence is a bit awkward. Please re-write for clarity.
The sentence was changed to improve clarity.
322-323 Is this because the cropping systems typically don't use any type of physical or mechanical control? You provided evidence earlier that some cultivation can be successful. This needs clarification.
By ‘physical weed control’ we refer to the methods discussed in the section with the same heading such as flaming and weed harrowing and as we concluded in that section these methods are not very effective against grass weeds like V. myuros.
330 Maybe it is because I am not very familiar with the cropping systems in Europe, but I am still not clear on how V. myuros vernalization affects management. Please elaborate.
In most part of Europe, we distinguish between summer and winter annual crops. Winter annual crops required vernalization to flower (move from the vegetative to the reproductive growth stage) and so does many weed species including V. myuros. By cultivating only summer annual crops the life cycle of V. myuros is disrupted and no seeds are produced and this will eventual empty the soil seed bank.
350-352 Elaborate. Is this because it allows different herbicides and different timing?
No, it is because of the effect of crop rotation on V. myuros seed production as explained above.